


# Automated avalanche hazard indication mapping on state wide scale

Yves Bühler[1,2], Peter Bebi[1,2], Marc Christen[1,2], Stefan Margreth[1], Lukas Stoffel[1], Andreas Stoffel[1,2], Christoph Marty[1], Gregor Schmucki[1,2], Andrin Caviezel[1,2], Roderick Kühne[3], Stephan Wohlwend[4], Perry Bartelt[1,2]

[1]WSL Institute for Snow and Avalanche Research SLF, Davos Dorf, 7260, Switzerland
[2]Climate Change, Extremes and Natural Hazards in Alpine Regions Research Center CERC, 7260 Davos Dorf, Switzerland
[3]Department of Forest and Natural Hazards AWN, Canton Grisons, Chur 7000, Switzerland
[4]Office for Civil Protection, Government of Liechtenstein, Vaduz 9490, Liechtenstein

*Correspondence to*: Yves Bühler (buehler@slf.ch)

## Abstract

Snow avalanche hazard mapping has a long tradition in the European Alps. Hazard maps delineate areas of potential avalanche danger and are only available for selected areas where people and significant infrastructure are endangered. They have been created over generations, at specific sites, mainly based on avalanche activity in the past. For a large part of the area (90% in the case of the Canton of Grisons) no maps are available. This is a problem when new territory with no or incomplete historical record is to be developed. It is an even larger problem when trying to predict the effects of climate change at the state scale where the historical record may no longer be valid. To close this gap, we develop an automated approach to generate spatial continuous hazard indication mapping based on a digital elevation model for the canton of Grisons (7105 km²) in the Swiss Alps. We calculate eight different scenarios with return periods ranging from frequent to very rare as well as with and without taking the protective effects of the forest into account. This approach combines the automated delineation of potential release areas, the calculation of release depths and the numerical simulation of the avalanche dynamics. This procedure can be applied worldwide, where high spatial resolution digital elevation models, detailed information on the forest and data on the snow climate are available, enabling reproducible hazard indication mapping also in regions where no avalanche hazard maps yet exist. This is invaluable for climate change studies. The simulation results are validated with official hazard maps, by assessments of avalanche experts and by existing avalanche cadastres derived from manual mapping and mapping based on satellite datasets. The results for the canton of Grisons are now operationally applied in the daily hazard assessment work of the authorities. Based on these experiences, the proposed approach can be applied for further mountain regions.

## 1 Introduction

Hazard maps are a key tool to cope with avalanche hazard in settled regions within alpine terrain, delineating areas with high risk, where buildings and infrastructure should not be erected (Rudolf-Miklau et al., 2014). In Switzerland and most other alpine countries, hazard maps are generated by avalanche experts for individual avalanche tracks, combining historical records, field investigations, terrain analysis, forest information and numerical modelling. Hazard maps proved to be an effective and reliable tool to prevent avalanche damage and victims (Margreth and Romang, 2010). However, hazard maps are only available for selected areas, mainly where infrastructure already exists and are very costly to elaborate in particular for large regions. For the canton of Grisons in Switzerland for example, only 10% of the alpine area are covered by hazard maps. If new buildings and infrastructure are planned, limited information on avalanche hazard is available in most areas. Furthermore, there are many mountain regions worldwide, where no hazard maps exist at all. To overcome this gap, we developed a computer based, automated procedure to generate reproducible avalanche hazard indication maps over large regions based on digital elevation model (DEM) data. In addition to the detailed hazard maps, hazard indication maps show the potential hazard areas outside the settlement area. These maps contain rough model-based estimates of the maximum hazard area affected in the case of an extreme event; however, they usually do not contain any information about the intensities that will occur.



Avalanche hazard indication mapping has been tested in different regions before. In 2004, the project SilvaProtect-CH started a first attempt to model avalanche hazard to assess the protective effects of the forest in Switzerland (BAFU, 2013; Gruber and Baltensweiler, 2004). This approach was based on a 25 m resolution DEM and the numerical avalanche simulation tool

AVAL-2D (Gruber and Bartelt, 2007). A generalized version of these results is still applied in Switzerland today resulting in hazard indication maps delineating the vast majority of alpine terrain a hazard area, lacking detailed information. This coarse level of detail is of limited use for most practical applications. Ghinoi and Chung (2005) developed a statistical model to asses avalanche susceptibility in the Italian Dolomites but without modelling avalanche runout. Barbolini et al. (2011) developed a semi-automated procedure to map avalanche hazard in two test sites in Italy based on the simple α runout model (McClung

and Mears, 1991) and statistical modelling (Barbolini and Keylock, 2002). Soteres et al. (2020) produced an avalanche susceptibility map of the Circo de Gredos in Spain applying GIS tools and collection of information from newspaper, field observations and feedback from backcountry users. They do not account for avalanche runout outside observed avalanche tracks. Horton et al. (2013) applied the hydrological flow model Flow-R for different processes such as rockfalls, debris flows and snow avalanches. The avalanche hazard mapping was calculated for an area of 25 km$^2$. Aydin and Eker (2017) modelled

avalanche hazard in the Bayburt region in Turkey based on a 10 m resolution DEM applying the Austrian Elba+ model, an improved version of the Elba model (Volk and Kleemayr, 1999). They further developed their approach for the Çaykara District (Aydin et al., 2021). Even though an avalanche runout model is applied, only one scenario is calculated. Larsen et al. (2020) developed an avalanche terrain exposure scale (ATES) for Norway, applying the raster based TauDEM-model. Issler (2020) developed NAKSIN, a new methodology for avalanche hazard indication maps in Norway. Choubin et al. (2019)

applied machine learning methods to predict avalanche hazard areas in the Karaj watershed in Iran. They focus on topographical and partly negligible land cover variables and do not apply an avalanche runout model. All of these recent approaches contain interesting components but are not able to produce reliable and validated avalanche runout modelling. However, this is key for hazard indication mapping within the settled areas of alpine valleys.

By developing a reliable and carefully validated algorithm to delineate potential avalanche release areas (PRA) (Maggioni and Gruber, 2003; Bühler et al., 2013), we initiated the possibility to apply the state-of-the-art avalanche dynamics model RAMMS (Christen et al., 2010b) to automatically generate hazard indication maps on a large scale (Bühler et al., 2018b) and to produce terrain classifications for backcountry touring (Harvey et al., 2018). Avalanche experts worldwide apply the RAMMS model since more than 10 years for hazard mapping and mitigation measure planning (Rudolf-Miklau et al., 2014) making it the most

applied, tested, validated and therefore trusted avalanche dynamics model for the simulation of dense flow avalanches. A main advantage of this approach is that it accounts for the standard return periods for hazard mapping applied in Switzerland and can therefore produce different hazard scenarios. We apply this newly developed automated hazard indication mapping tool to the canton of Grisons in Switzerland, validate the results with existing hazard maps, assessments of avalanche experts, historical avalanche cadastres and avalanche records mapped by satellites. Then we discuss the strength and weaknesses of

this approach and provide an outlook on future applications and potential improvements.

## 2   Canton Grison

The Swiss canton of Grisons with a total area of 7105 km$^2$ is the largest state of Switzerland (total area 41'285 km$^2$). 100 % of its area is alpine terrain, ranging from Piz Bernina (4049 m a.s.l.) down to the river Moësa in the Misox valley (252 m a.s.l.) at the border to the canton Ticino (Figure 1). With a permanent population of roughly 200'000 persons its population density

is the thinnest in Switzerland (28 persons per km$^2$; average in Switzerland: 215 persons per km$^2$). However, during winter season roughly 2.5 million tourists (overnight stays) visit the more than 40 ski resorts in the canton every year. Some of these



ski resorts are world-famous, for example St. Moritz, Davos-Klosters, Arosa-Lenzerheide or Flims-Laax. To provide and secure the infrastructure for all these people in high alpine terrain is a challenging task.

5   The extreme relief causes frequent gravitational mass movements such as snow avalanches, rockfalls, debris flows and landslides. An severe event took place at the Pizzo Cengalo in 2017 where roughly 3 million m³ of rock caused a large rock avalanche, killing eight people and causing large debris flows reaching the village of Bondo at the valley bottom (Walter et al., 2020). Extreme avalanche winters such as 1999 (SLF, 2000) and again 2018 and 2019 (Bühler et al., 2019; Bründl et al., 2019; Zweifel et al., 2019) caused damage but thanks to well established protection strategies (Bründl and Margreth, 2021) only few victims.

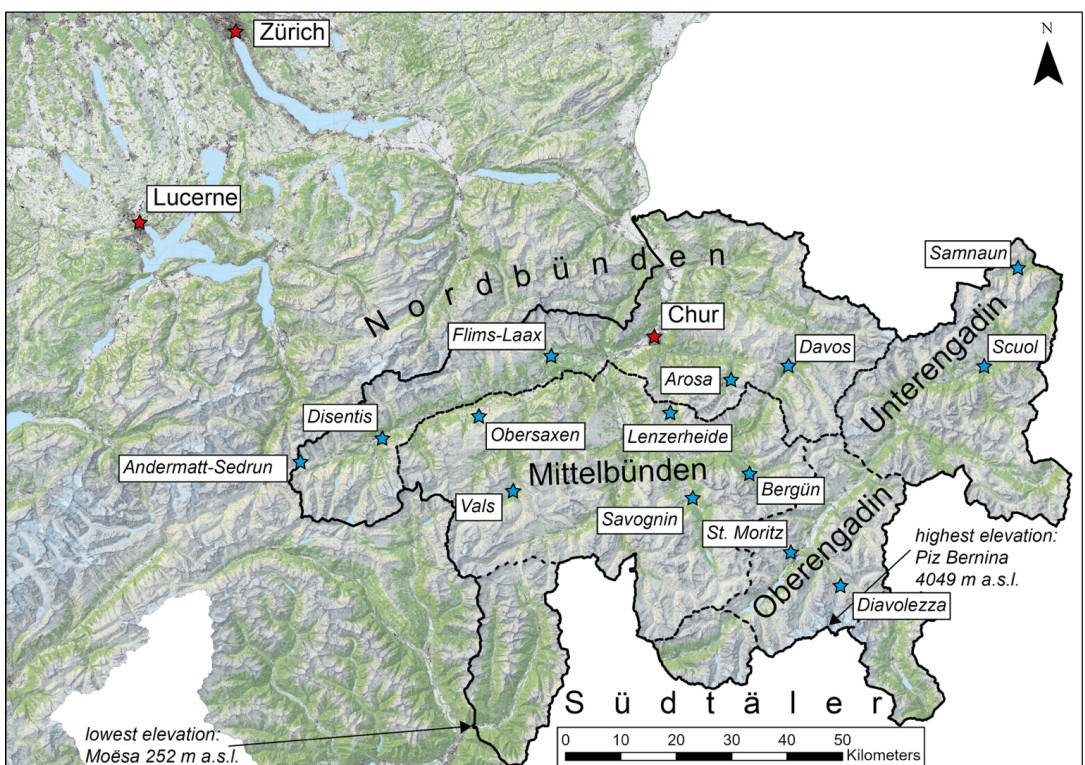

**Figure 1: Map of Eastern Switzerland visualizing the extent of the canton Grisons (black outline), the major towns (red stars) and the largest ski-resorts (blue stars) as well as the climatic regions (dashed lines) applied in section 3.4 (map source: Swiss Federal Office of Topography)**

The Department for Forest and Natural Hazards AWN of the canton Grison developed hazard maps for the settled regions in

15   collaboration with local engineering offices. However, these maps only exist within predefined areas, where buildings and infrastructure are at immanent risk. This is the case for approximately 10 % of the area of the canton Grisons. For the remaining 90 % of the area, no hazard maps exist. If new infrastructure is planned in areas outside the hazard maps, if existing settlements grow into such areas or a detailed assessment of the protective effects of forests is necessary, the basic information allowing for risk assessment have to be generated. Therefore, the AWN assigned the WSL Institute for Snow and Avalanche Research

20   SLF to develop an approach to fill the blind spots and to generate a hazard indication maps for snow avalanches for the entire area of the canton. The approach should be automated and build on available geodata.



## 3    Methods

### 3.1    Digital Terrain Model DTM and scenario setting

The base for the delineation of the potential release areas (PRA, chapter 3.3) and the numerical avalanche simulations (chapter 3.5) is the digital elevation model DEM provided by swisstopo. The applied digital terrain model DTM swissALTI[3D], representing the bare earth surface without vegetation and buildings, is available for the entire area of Switzerland with a
spatial resolution of 2 m (swisstopo, 2018). It is produced based on airborne laser scanning (below 2000 m a.s.l.) and airborne photogrammetry (above 2000 m a.s.l.) data and has a nominal accuracy of 0.5 m x,y and z for the lower part and 1 – 3 m for the higher part of the DTM. Due to the smoothing effect of the snow cover on the terrain, the original DTM was resampled to a resolution of 5 m (Bühler et al., 2018b; Bühler et al., 2018a). Based on this DTM product, the derivatives slope, curvature, 10    aspect and ruggedness are calculated with ArcGIS Pro tools (ESRI).

Based on the standard procedure for snow avalanche hazard mapping in Switzerland defined by the Federal Office for the Environment FOEN in collaboration with SLF, we simulate the four return periods 10, 30, 100 and 300 years. To assess the protective effects of the forest, we simulate these scenarios once taking the forest effects into account for the PRA (For) and
once without taking it into account (NoFor). This results in 8 different scenarios.

### 3.2    Forest information

In order to define forests with a potentially protective effect against avalanches, we deduce objective criteria for potential avalanche releases in forested terrain and implement them within a GIS-approach ((Bebi et al., 2021), Figure 2). We therefore statistically analyse data on forest structure and topography of 150 avalanche releases in forested terrain (Schneebeli and
Meyer-Grass, 1993) and use the modelling approach of Bebi et al. (2001) to deduce threshold values for relevant explanatory variables. The resulting logistic regression model is reduced to the variables "slope inclination", "percentage of crown cover" and "gap width" and allows us to deduce an "avalanche disposition" between 0 (no disposition) and 100% (very high avalanche release probability) to each forested raster cell of 5 x 5 m. To minimize the calculation time, we applied the model only within a forest mask, defined by forested areas according to the Federal Office of Topography (swisstopo) and the Swiss National
Forest Inventory, NFI. We define different threshold values for tree heights to assign forest gaps ("gap-threshold") and forest cover ("forest cover-threshold") for the two different avalanche scenarios (frequent and very rare). The two scenarios apply regional differences in snow depth distribution according to Margreth (2007). In order to account for increased protection forest capacities of high terrain roughness compared to smooth surfaces, we delineate areas with a high surface roughness and assigned a higher protection forest index to areas with high roughness, which do not show lateral convex curvature (Brožová
et al., 2021). Likewise, we account for reduced protection forest capacities of shrubs (eg. Green Alder, *Alnus viridis* or the shrub form of Mountain pine*, Pinus mugo turra subsp. mugo*) compared to trees with erect growth forms of the same height. We consider the scrub forest area layer according to Weber et al. (2020) and assign a lower protection forest index to areas covered by shrubs (Figure 2). Based on the resulting "protection forest index" we then deduce threshold values defining a sufficient protection forest index for frequent (10-30 years return period), and very rare (100-300 years return period) scenarios.
We finally correct the protection forest thresholds and other threshold values in an iterative process after validating avalanche simulations with former avalanche events and after discussing different scenarios together with the responsible regional natural hazard experts (Bebi et al., 2021).


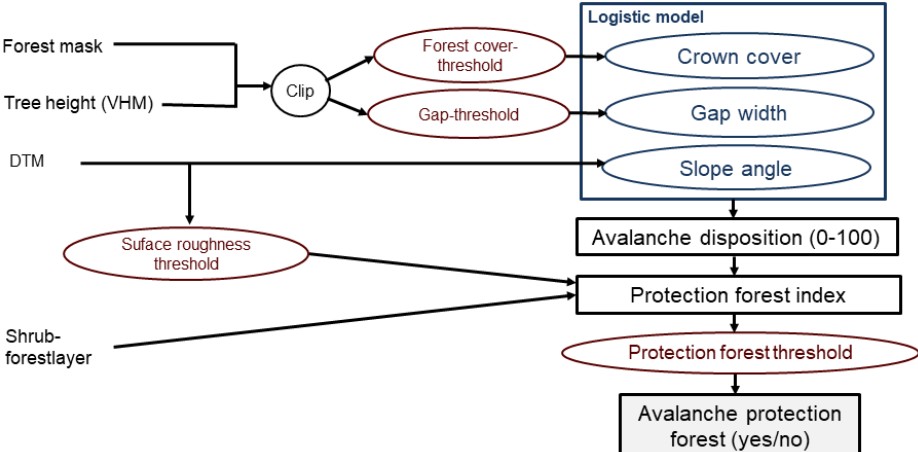

**Figure 2: Schematic structure of the model for calculating the spatial extent of potential avalanche protection forest, simplified according to (Bebi et al., 2021).**

### 3.3 Potential Release Area (PRA) identification

The detailed approach to automatically identify and delineate the potential release areas (PRA), combining GIS terrain analysis with object based image analysis (OBIA) is described in Bühler et al. (2018b) in detail. The only adaption is that the fold (a derivative of the curvature) is replaced by the curvature itself. The main input datasets are the binary forest information (chapter 3) and the digital terrain model (DTM, chapter 3). To efficiently process the large area, the different steps are programmed in python scripts and are run by ArcGIS Pro. To avoid PRA with very low release probability, polygons with areas less than 1000

$m^2$, a mean slope angle of less than 28° or with a mean elevation of less than 600 m a.s.l. are excluded. These thresholds were elaborated in collaboration with the cantonal experts.

### 3.4 Release depth d0

The calculation of the release depth d0, the SLF standard procedure (Salm et al., 1990) is carried out as follows: For each PRA polygon, the specific snow climate region is assigned (Figure 1). Based on the long term o data series from the SLF observer

network (https://www.slf.ch/en/avalanche-bulletin-and-snow-situation/measured-values/description-of-manual-measuring-sites.html, last access 14. January 2022), the annual maxima of snow depth increase (in cm) within 3 days (ΔHS3, Figure 3) are then calculated and the return period (e.g., for a 10-year event) is determined. This return level is adjusted to the height of the respective polygon using a gradient of 5 cm per 100 meters elevation change. Next, a flat-rate drifting snow surcharge is added. Below 1000 m no surcharge is added, above 2000 m the full surcharge (300y: 0.5 m. 100y: 0.3 m, 30y: 0.2 m and 10y:

0.1 m) is added, the local topography and snow drift situation was not taken into account. In between, linear interpolation is used to give more weight to the altitude dependence. Finally, the slope gradient is considered by assuming that less snow can be accumulated in steep areas than in flat areas. We improved the standard procedure by replacing the Gumbel distribution for the return level estimation with the Generalized Extreme Value (GEV) distribution. The GEV allows for three different types of distributions (one of which is the Gumbel distribution), which more closely matches the possible distribution of the

measured data. On the other hand, we determined the return values of ΔHS3 per climate region and not per station. The reason for this is the large estimation uncertainties of return values for the individual stations.

To derive the ΔHS3 values for the five different snow climate regions, the daily data of 44 long-term, manual SLF and MeteoSwiss snow measuring stations between 1000 and 2600 m are used. The minimum measurement duration is 25 years,

the median value of the measurement duration of all stations is 65 years. We decide to divide the canton into the following


regions with respect to different snow climates: *Oberengadin, Unterengadin, Mittelbünden, Nordbünden, Südtäler* (Figure 1). However, it must be taken into account that especially the determination of the return values of 100- and 300-year events reveal such large uncertainties for all stations (Figure 3) that a clear distinction between the regions is hardly possible. The original goal was to determine the elevation gradients per climate region. A separate study of the same 44 stations demonstrate

that the gradients strongly depend on the individual heavy snowfall events and on the selection of stations. It is therefore decided to continue with the old SLF standard gradient of ± 5 cm per 100 vertical meters. This means that the return values and their uncertainty for the calculation of the release depth are determined as a function of the elevation and the climatic region.

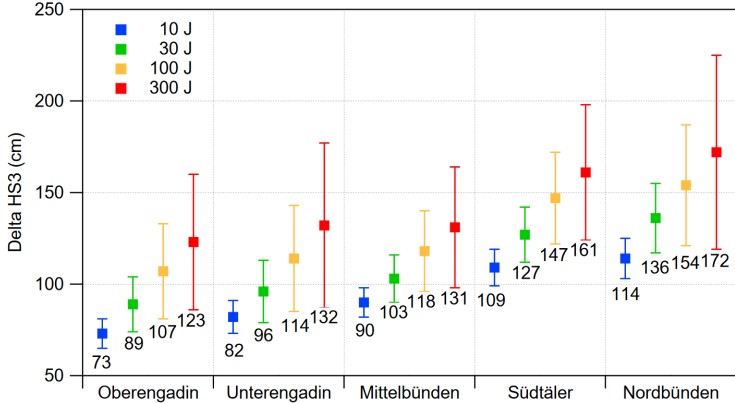

**Figure 3: ΔHS3 values and associated uncertainties for the five different climatic regions of Grisons and the four scenarios normalized to an elevation of 2000 m a.s.l.**

### 3.5 Numerical avalanche simulations with RAMMS::LSHIM (Large Scale Hazard Indication Mapping)

The PRA (chapter 3.3) for the eight scenarios (10y frequent, 30y average, 100y rare and 300y very rare, with and once without taking the protective effects of the forest into account) are then loaded with the corresponding d0 (chapter 3.4) based on their

location within one of the five climatic regions (Figure 3). Based on the size of the avalanche release area and d0, the PRA are classified into the volume classes *large* (> 60'000 m$^3$), *medium* (25'000 – 60'000 m$^3$), *small* (5'000 – 25'000 m$^3$) and *tiny* (< 5'000 m$^3$) to define the RAMMS model friction parameters μ (basal friction) and ξ (turbulent friction) according to the RAMMS standard modelling workflow (Christen et al., 2010b), which is applied in Switzerland for hazard mapping. Then the calculation domains are delineated for every individual PRA. Due to the large study area, a calculation grid of 10 m spatial

resolution is applied for the RAMMS::LSIHM calculations. The parameters setting is based upon the standard approach for avalanche hazard mapping in Switzerland. Curvature effects for the friction are considered, the cohesion is set to 50 Pa and the stop criterium is set to 5 % of the total momentum. Based on these input datasets, all individual PRA are simulated with RAMMS::LSHIM (Bühler et al., 2018a) in sequence as individual avalanches, not interacting with each other.

The output is then combined to raster maps of the maximum pressure, velocity and flow depth per cell. The outlines of the avalanches are exported as polygons containing the area with a maximum flow depth larger than 0.25 m and a maximum velocity larger than 1 ms$^{-2}$. For the scenarios taking forest into account (For), the forest layer (section 3.2) is additionally considered for the automatic definition of the friction values. For the PRA in the volume classes *small* and *tiny*, the turbulent friction ξ is set to 400 ms$^{-2}$, braking the velocity of the avalanche (Christen et al., 2010b). For the volume classes *large* and

*medium* this braking effect is neglected because a destruction of the forest is assumed. We perform these calculations on a DALCO Rackmount Server with 96 cores and 384 GB RAM. The calculation of one scenario takes about 24 hours for approximately 200'000 simulations.



### 3.6 Assumptions and Limitations

The presented approach automatically generates the potential release areas PRA, defines the release volumes d0 and simulates the avalanche runout, impact pressure, flow velocity and flow depth. To be able to automate this process, several important assumptions have to be made:

- The simulations are based on up-to-date digital elevation model data and the recent status of the forest. However, the terrain characteristics and in particular the forest structure can locally change (Bebi et al., 2009), impacting the avalanche hazard. Snow cover or avalanche deposits are not considered.

- The hazard indication maps do not take release probabilities into account. The basic assumption is that all potential release areas, derived with the PRA algorithm (section 3.3) will release and produce an avalanche. The return periods are only linked to the forest layer (frequent and very rare, section 3.2), the release depth d0 (section 3.4) and the friction values of the RAMMS model (section 3.5).

- Only the dense flowing part of dry avalanches is simulated. Powder snow avalanches or wet snow avalanches are not considered. Also, the occurrence of several subsequent avalanches within tracks as well as the interaction of different avalanches and snow entrainment is neglected.

- Mitigation measures within avalanche release zones are not taken into account, as a reliable GIS layer containing the necessary information is not yet available for the canton of Grisons yet. However, for the automated hazard indication mapping of Liechtenstein, where such a layer was available, the mitigation measures were considered. The effect of the supporting structures was checked by an expert, in particular on the basis of the height of the structure, the type of construction and the arrangement. In those areas of supporting structures that were found to be fully effective, no release areas were identified. The mitigation measures within the avalanche tracks, such as deflection and catching dams, are only applied if they are well depicted in the DTM data. This is one of the reasons for discrepancies to the official hazard maps (section 5.1).

- Not all PRA present in reality can be captured with the presented approach. For example, PRA caused by cornice failure or initiated in extremely steep rock faces are neglected. On the other hand, even the runout of the 300y scenarios do not completely cover all area that could be endangered by an avalanche in very rare scenarios. In particular where very large bowls are present, that could release as a whole in a very rare case, the release volume can be underestimated. Yet the reach of powder clouds and the breakout of side arms of wet snow avalanches can't be simulated.

These points need to be considered for the interpretation of the results and the application of the final products. Nevertheless, the generated maps are very valuable for numerous practical applications (section 6).

## 4 Results

### 4.1 Forest layers

Our automatized approach allowed us to define forests with potentially protective effects against avalanches for rare to very rare (100-300 years) avalanche events and for relatively frequent (10-30 years) events (Fig. 4). A total area of 1868 km² (71
% of the forested area in Graubünden) was assigned a protective effect against rare and very rare avalanche events. These forest areas are mainly characterized by a relatively dense to moderately dense forest cover of trees that clearly exceed the height of the area-specific extreme snow height. As slope angle is an important factor in the underlying logistic model, 37 % of these forests are on terrain with a steepness of < 35 °. Forests with a potentially protective effect against relatively frequent avalanches cover a total area of 2088 km² (79 % of the total forest cover). In addition to forests with an assigned protection
effect against rare and very rare avalanches, this category contains a larger variety of more open and less tall trees in a steeper



environment. Typical forests with an assigned effect against frequent but not against rare or very rare avalanche events are found in higher elevations close to treeline or in disturbed forests (Fig. 3).

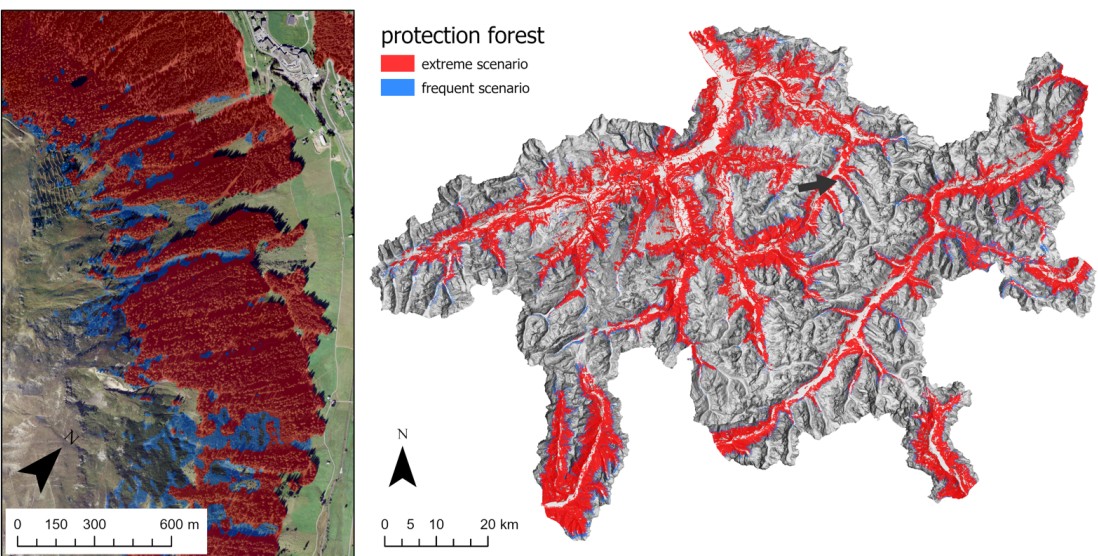

5   **Figure 4: Protection forest map for the Canton Graubünden for a frequent (10-30 years) avalanche scenario (corresponding to avalanche events with a 10 – 30-year return period, displayed in blue and red combined) and for an extreme snow cover scenario (corresponding to a ca. 100 – 300-year avalanche event, displayed in red). Orthophoto source: Swiss Federal Office of Topography.**

## 4.2 Release areas

In total, nearly 2 million individual avalanches are simulated. Figure 5 and Figure 6 show the results of the PRA delineation
10  (section 3.3) for all scenarios in an area of approximately of 4 km$^2$. The size of the PRA is increased with increasing return period. The forest, where present, prevents the formation of a release area and therefore reduced the PRA by roughly one third (Table 1). For the 300yFor scenario, 29 % of the area of the entire canton of Grisons (7105 km$^2$) is delineated as PRA. For the scenario 300yNoFor it is already 43 %. For the 10y scenarios it is 22 % (For) and 35 % (NoFor). These numbers illustrate the steep topography and the high protective effects of forests in the canton of Grisons for the prevention of avalanche release.



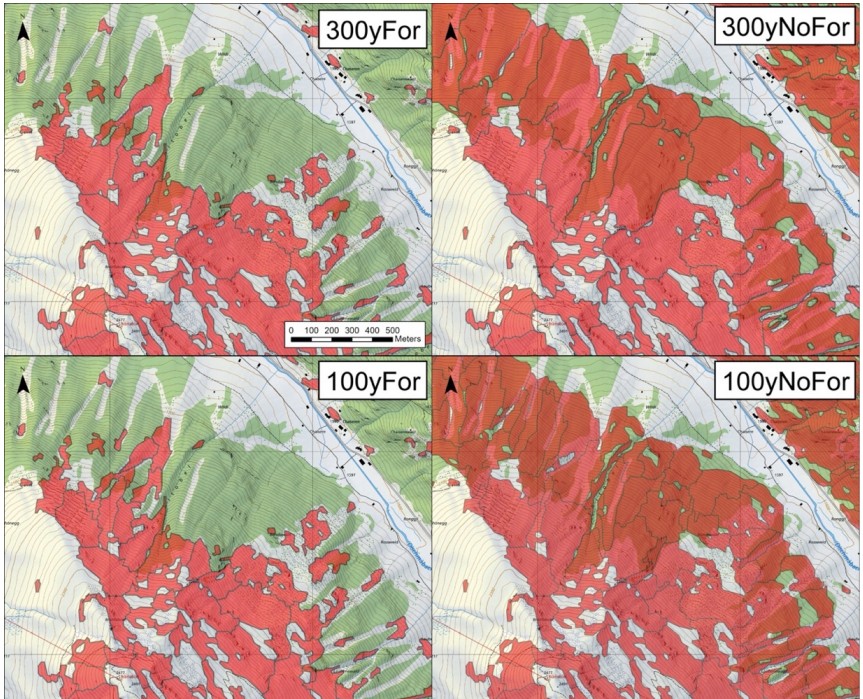

**Figure 5: PRA delineation at the Duchli slope at the beginning of the Dischma valley close to Davos for the two rare scenarios (300y and 100y) with (For) and without (NoFor) taking the protective effects of the forest into account (map source: Swiss Federal Office of Topography).**

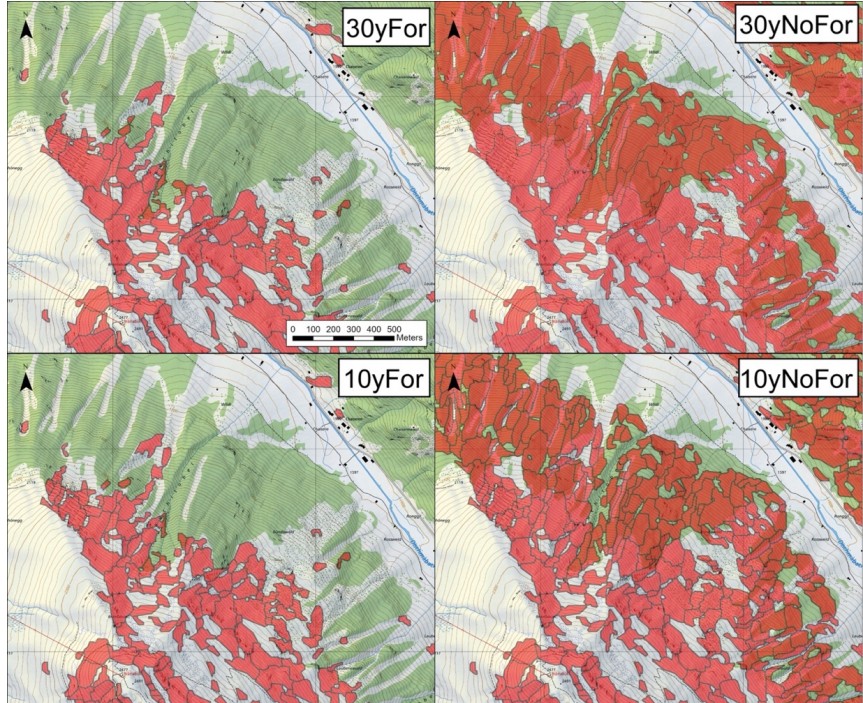

**Figure 6: PRA delineation at the Duchli slope at the beginning of the Dischma valley close to Davos for the two frequent scenarios (30y and 10y) with (For) and without (NoFor) taking the protective effects of the forest into account (map source: Swiss Federal Office of Topography).**





Table 1 lists the statistical overview on the resulting PRAs for the individual scenarios. The combination of reduced area with reduced release depth (d0, section 3.4) results in reduced release volume of the PRA for the more frequent scenarios. The forested areas (forest boundaries at ~ 2100 m a.sl. in the northern and ~2300 m a.sl. in the southern regions), cause a reduction

5   of the average release elevation of ca. 150 m (300y) – 300 m (10y). By not taking the protective effects of the forest into account, 20 % (300y) up to 35 % (10y) more PRA are delineated in forested terrain. Because these regions are at lower elevation below the treeline, the average d0 and slope angle is higher for the scenarios taking the forest into account (FOR). The average d0 varies between 1.29 m (300yFor) and 0.71 m (10yFor).

10   **Table 1: Statistical overview on the PRA for the eight different scenarios over the entire canton of Grisons**

| Szenario name | Number of PRA | Covered area [km²] | Ø elevation [m a.s.l.] | Ø slope angle [°] | Ø release depth d0 [m] | Ø plan view area [m²] | Ø Volume [m³] |
|---|---|---|---|---|---|---|---|
| **300yFor** | 88'589 | 2'056 | 2'199 | 35.93 | 1.29 | 23'211 | 37'440 |
| **300yNoFor** | 108'551 | 3'075 | 2'040 | 35.60 | 1.23 | 28'336 | 41'486 |
| **100yFor** | 159'794 | 2'047 | 2'280 | 36.35 | 1.08 | 12'815 | 17'153 |
| **100yNoFor** | 216'709 | 3'061 | 2'059 | 36.08 | 1.00 | 14'127 | 17'144 |
| **30yFor** | 196'940 | 1'643 | 2'352 | 37.75 | 0.88 | 8'341 | 9'256 |
| **30yNoFor** | 295'168 | 2'634 | 2'067 | 37.47 | 0.79 | 8'924 | 8'799 |
| **10yFor** | 366'150 | 1'574 | 2'380 | 38.15 | 0.71 | 4'298 | 3'927 |
| **10yNoFor** | 563'460 | 2'521 | 2'078 | 37.89 | 0.63 | 4'475 | 3'582 |
| **Over all** | 1'995'361 | - | 2'182 | 36.90 | 0.95 | 13'066 | 17'348 |

### 4.3 Simulation results

To provide an overview on the simulation results for the entire canton of Grisons Figure 7 depicts the endangered areas for the most rare (300yFor) and the most frequent (10yFor) scenario. Because the canton of Grisons is situated entirely within the Alps, a large portion of the area is endangered by avalanches. Even in the most frequent scenario, a considerable part of the

15   canton could be hit by avalanches with impact pressures of more than 30 kPa. The consequence is that a large part of the canton of Grisons, mostly at higher altitudes, would be classified as a red hazard zone. As only about 10 % of the area are of the canton Grisons are covered by these hazard maps, mostly within isolated areas where infrastructure is at risk, the spatial continuous indication maps presented in this paper are of high value. For 90 % of the area of the canton, no detailed information on avalanche hazard existed before.

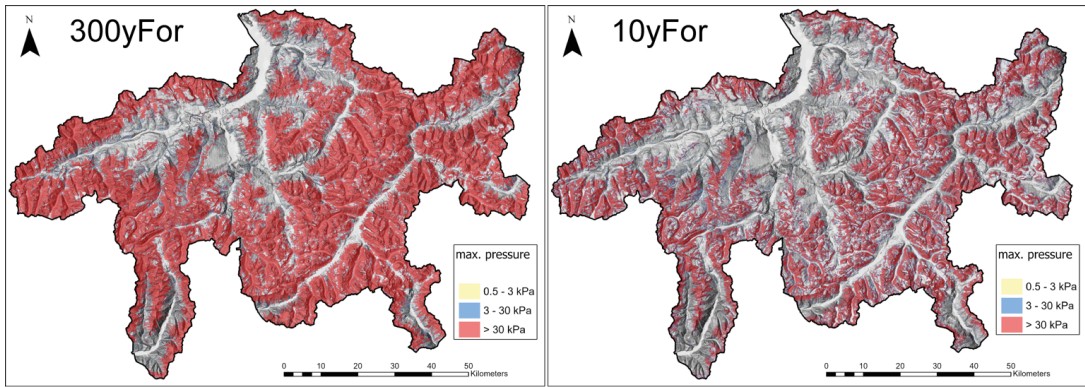

**Figure 7: Simulation result for the entire canton of Grisons for the most rare (300y, left) and the most frequent (10y, right) scenario, taking the protective effects of the forest into account (hillshade source: Swiss Federal Office of Topography).**


To set the affected area in relation to the total area of the canton Table 2 gives a summary of the different scenarios. Without the protective effects of the forest, 5'972 km² or 84 % of the total area would be endangered during a very rare avalanche period (300yNoFor). The forest reduces this endangerment by 17 % down to 67 % (300yFor). The reduction of the endangered area is increasing to 19 % for the frequent scenarios (10y and 30y). This is a lot taking into account that only 1'852 km² or 26 % of the cantonal area are covered by high timber forest, highlighting the high protective value of the forest. Even within the most frequent scenario (10yFor) and taking the protective effects of the forest into account, still 47 % of the area of the canton are endangered by avalanches.

**Table 2: Statistical overview on the simulation results for the eight different scenarios over the entire canton of Grisons**

| Szenario name | Affected area [km²] | Affected area compared to the total area of the canton (100 % = 7105 km²) | Release Areas PRA [km²] | Release areas PRA compared to the total area of the canton (100 % = 7105 km²) |
|---|---|---|---|---|
| 300yFor | 4'791 | 67 % | 2'056 | 29 % |
| 300yNoFor | 5'972 | 84 % | 3'075 | 43 % |
| 100yFor | 4'388 | 62 % | 2'047 | 29 % |
| 100yNoFor | 5'625 | 79 % | 3'061 | 43 % |
| 30yFor | 3'670 | 52 % | 1'643 | 23 % |
| 30yNoFor | 5'063 | 71 % | 2'634 | 37 % |
| 10yFor | 3'339 | 47 % | 1'574 | 22 % |
| 10yNoFor | 4'661 | 66 % | 2'521 | 35 % |

To illustrate the level of detail contained in the simulation output, Figure 8 and Figure 9 show the simulated maximal impact pressure for the Duchli slopes in Davos for all scenarios. The corresponding release areas are depicted in Figure 5 and Figure 6. This very steep slope, where frequent avalanche activity is observed, illustrates the differences in impact pressure and runout distance between the scenarios. In the avalanche tracks, a pressure of 300 kPa is exceeded in many places in nearly all scenarios. The effect of the forest is not very obvious for the rare scenarios (Figure 8) because the large PRAs are located high above treeline (Figure 5) and the forest is assumed to be destroyed by the high impact pressure of the resulting avalanches. For the frequent scenarios however (Figure 9), the forest slows down and stops the smaller avalanches. Only avalanches in the forest free tracks, where avalanches occur practically every winter, reach the valley bottom. The visualization of the maximum impact pressures allows for a detailed investigation of the endangered terrain.

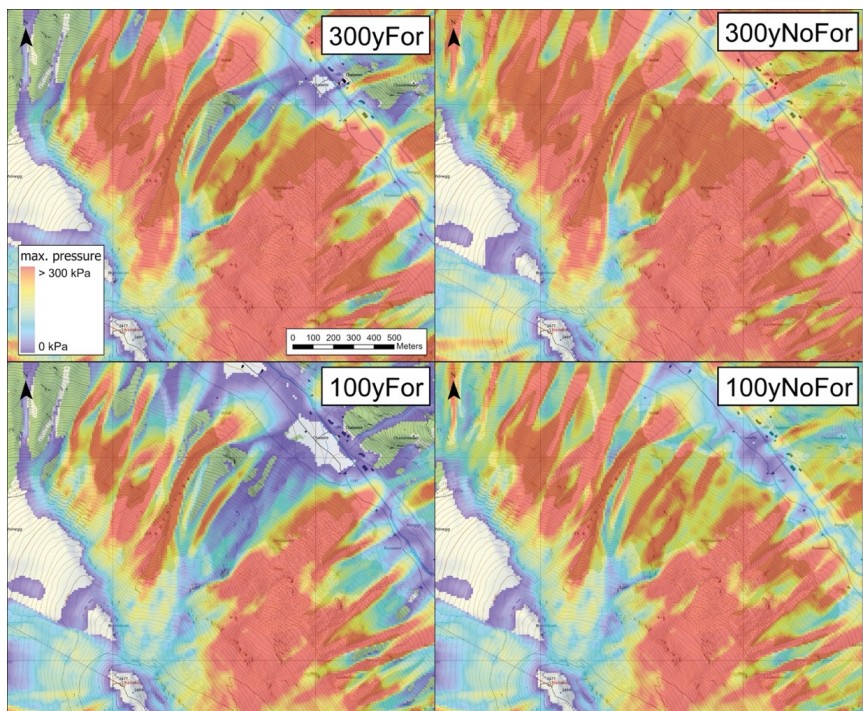

**Figure 8: Maximum pressure of the simulated avalanches for the two rare scenarios for the subsection Duchli. The corresponding PRAs are depicted in Figure 5 (map source: Swiss Federal Office of Topography).**

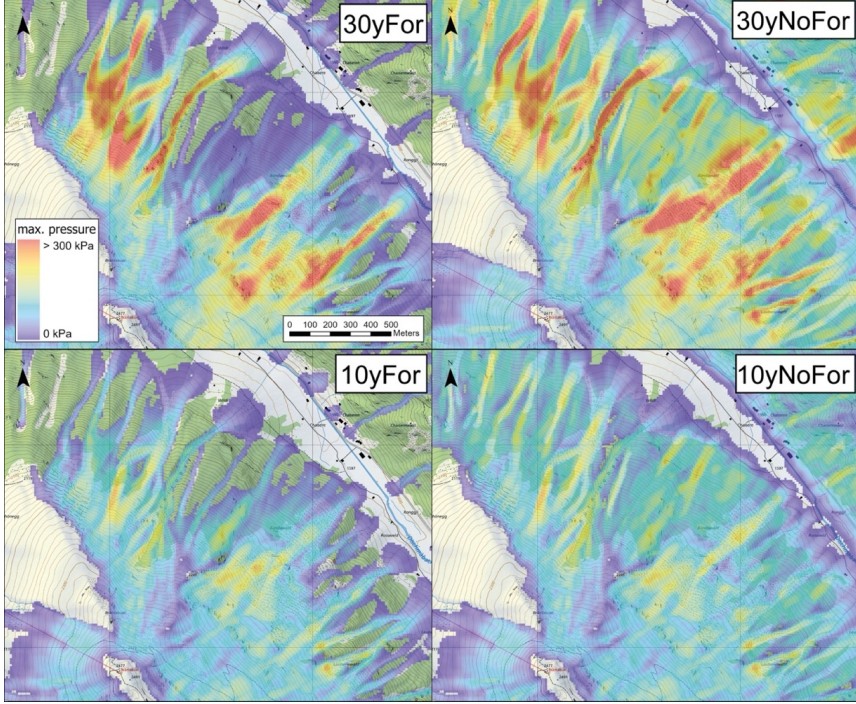

**Figure 9: Maximum pressure of the simulation results for the two frequent scenarios for the subsection Duchli. The corresponding PRAs are depicted Figure 6 (map source: Swiss Federal Office of Topography).**


## 5   Validation

A meaningful validation of the calculated hazard indication maps is very challenging. Spatial continuous and reliable information on avalanche occurrence with a time span of decades to centuries is not available. To validate our product, we therefore use the best information available:

- Hazard maps of the canton of Grisons (https://edit.geo.gr.ch/theme/Naturgefahrenkarte). These legally binding hazard maps are produced by avalanche experts for individual avalanche tracks by taking different sources into account. The most important sources are avalanche cadastres, terrain analysis, climatic situation, field investigations, numerical simulations and expert experience (Rudolf-Miklau et al., 2014).

- The event register of the canton of Grison. In Switzerland, the cantons are obliged to keep an event register. Some of
the documented events date back to the 17th century.

- SLF avalanche cadastre, which has a focus on the regions Davos and Zuoz in Grisons (Schweizer et al., 2020; Hafner et al., 2021).

- Avalanches mapped based on optical and radar satellite imagery for two large avalanche periods in January 2018 and January 2019 (Bühler et al., 2019; Bründl et al., 2019; Zweifel et al., 2019; Hafner et al., 2021; Bühler et al., 2021).

- Qualitative comparison with hazard assessments of avalanche experts from SLF, the canton of Grisons and Liechtenstein.

However, the hazard maps are only available within settled areas where the community has initiated a hazard assessment (about 10 % of the cantonal area). The coverage and quality of the avalanche cadastre is dependent on the individual observers, is incomplete and contains mapping errors (Bühler et al., 2018b). The satellite mapping only covers two recent, large avalanche
periods (Hafner et al., 2021; Bühler et al., 2021). Despite these limitations these datasets were the best available to validate our product.

### 5.1   Comparison to official hazard maps

The most reliable validation is the direct comparison of the 300yFor scenario with the existing hazard maps within settled areas (Margreth, 2019). Not yet all hazard maps are generated with state-of-the-art methodology. Furthermore, the hazard
maps partially take mitigation measures into account. Together with avalanche experts from the canton and from SLF, the simulation results were compared to existing hazard maps at different locations within the canton. The region of Davos is selected to illustrate such a comparison (Figure 10).

Davos, 1560 m a.s.l., is the second largest city of the canton (~12'000 inhabitants) after the capital Chur and has a well-
documented avalanche history reaching back to the settlement of the Walser tribe in the 13th century. As domicile of the WSL Institute for Snow and Avalanche Research SLF, founded in 1936, this region has developed one of the most reliable avalanche cadastre and maintains a rigorous mapping of recent avalanche events (Hafner et al., 2021). The hazard maps of the area are an important tool for hazard management and were revised several times since the first version.

The difference of the spatial coverage between the hazard maps and the presented simulations is well visible in Figure 10. The runout of the Salezer avalanche (1 in Figure 10) agrees well with the simulation results. However, in the hazard map the blue and yellow zone reach about 200 meters further to the east than in the simulation. This is because the runout of the powder snow avalanche is much longer compared to the runout of large dry dense avalanches. This effect is not simulated in the presented approach (see section 3.6). The large avalanche was artificially triggered in January 2019 and the estimated return
period is about 10 years. The large powder avalanche developed a runout as predicted for the 100y return period in the indication map. The runout of the Lusi avalanche (2 in Figure 10) agrees very well with the simulation and observations. The





Schiawang avalanche (3 in Figure 10) shows major differences. While the hazard zones end in the area of the topmost buildings, the simulated hazard zones run further down, impacting many buildings. This is because there are snow supporting structures in the release zone of this avalanche. They are considered for the hazard map but not for the simulations (see section 3.6). However, when the supporting structures would not be functional anymore, the situation could get as depicted in the simulation. In February 1817, long before the first protection measures in the release area of the Schiawang avalanche were built, the observed runout distance was even longer than in the simulated hazard zones. The Arelen avalanche (4 in Figure 10) has released several times in the last decades (Christen et al., 2010a) and reached very long runouts as indicated by the hazard map. However, a few years ago a new protection dam against debris flows was erected in the channel. Finally, the Dorfbach avalanche (5 in Figure 10) is rather well modelled. The Dorfbach avalanche was very large in 1968 and 1609. The extent of these avalanches is covered by the official hazard map. The simulated hazard zones are somewhat longer. A reason might be that the braking effect of existing building is not considered in the simulations. The overall impression of the avalanche experts examining the results is that taking the assumptions and limitations described in section 3.6 into account, the simulation results agree rather well with the official hazard maps.

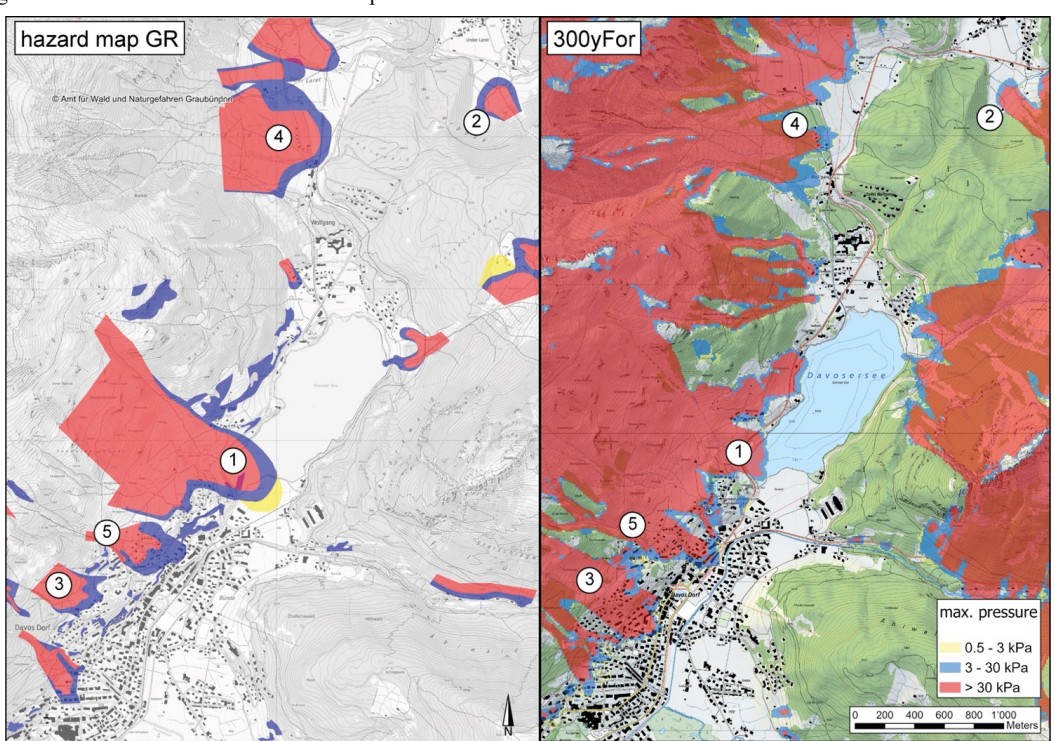

**Figure 10: Comparison of the official hazard map (map.geo.gr.ch, left) with the simulation results of the scenario 300yFor (right). The numbers indicate the location of selected avalanche tracks discussed in this section (map source: Swiss Federal Office of Topography).**

## 5.2 Comparison to the SLF avalanche cadastre and recent events mapped by satellite

During January 2018 and again in January 2019, Switzerland experienced extreme snowfall events (Bründl et al., 2019; Zweifel et al., 2019). For the first time since 1999 the avalanche warning reached level five, the highest level of the danger scale. These extreme storm cycle resulted in numerous very large wet (2018) and dry (2019) snow avalanches with partly long runouts. To accurately map these avalanches, optical and radar satellite data was applied to cover roughly 12'000 km$^2$ of the Swiss Alps and large parts of the canton of Grisons (Bühler et al., 2019; Leinss et al., 2020; Hafner et al., 2021). In total approximately 25'000 individual avalanche outlines were mapped using satellite data, 6'737 of them within the canton of




Grisons (Bühler et al., 2021). The main uncertainties occur in areas, which were in cast shadow in the optical imagery (Hafner et al., 2021). Additionally, the SLF and its observer network map avalanches for the SLF avalanche database (Schweizer et al., 2020). This dataset contains mapping errors mainly due to inaccurate or even false localization of the avalanche events in the topographic maps by the persons mapping the avalanche events. Even though all three of these datasets are affected by

specific mapping problems and are not complete, they still allow for a quantitative validation of the area affected by avalanches. By comparing how much area is affected within the reference datasets (SLF database and satellite mapping) but not in the 300yFor simulation results (Table 3), it can be assessed how much endangered area is not captured by the simulations results. Because it cannot be stated that areas where no avalanche is captured in the SLF database nor has occurred in January 2018 and 2019 are safe from avalanches, the area that is falsely affected in the simulation result cannot be assessed with these

datasets. Comparing the simulations without taking the protective effects of the forest into account would not make sense, as the area would be massively overestimated. For the scenario 300yFor, only 1.4 % of the area is not captured as endangered by our simulations (Table 3). As the reference datasets contain mapping errors, a part of this area could be caused by these mapping errors and not by an insufficient performance of the simulations.

**Table 3: Avalanche affected area in the reference datasets (SLF database and satellite mapping) within the canton of Grisons, which are not affected in the simulation results in comparison to the total affected area (%). An example for visualisation of a subset regions is given in Figure 11.**

| Scenario name | Satellite mapping January 2018 [m$^2$] | Satellite mapping January 2019 [m$^2$] | SLF avalanche cadastre [m$^2$] | Total [m$^2$] |
|---|---|---|---|---|
| Total area affected | 164'908'439 | 170'910'105 | 195'931'333 | 450'355'442 |
| 300yFor | 1'810'610 | 1'391'908 | 3'359'943 | 6'190'520 |
|  | 1.09 % | 0.8 % % | 1.7 % | 1.4 % |
| 100yFor | 2'600'451 | 2'058'328 | 5'834'304 | 9'709'646 |
|  | 1.6 % | 1.2 % | 3.0 % | 2.2 % |
| 30yFor | 4'857'589 | 4'654'960 | 14'360'605 | 21'877'178 |
|  | 2.9 % | 2.7 % | 7.33 % | 4.9 % |
| 10yFor | 8'702'270 | 9'177'882 | 22'741'031 | 36'779'433 |
|  | 5.3 % | 5.4 % | 11.6 % | 8.17 % |

To visualize the applied reference datasets and the comparison with the 300yFor scenario, Figure 11 shows a subset of the

comparison south of Davos. The avalanche events recorded in the SLF database that released in the forest (south-eastern area in Figure 11) are not all captured in the simulations. This could be caused by changes in the forest structure during the past decades. As the SLF database contains avalanche events back to the year 1888 with continuous mapping since 1951, the forest and its protective effects may have changed significantly since back then. Figure 11 demonstrated that a large part of the area has no observation of past avalanche events but still is endangered according to the simulations. This is even more distinct in

regions which are outside of the focus area of the SLF database (Schaer, 1995).

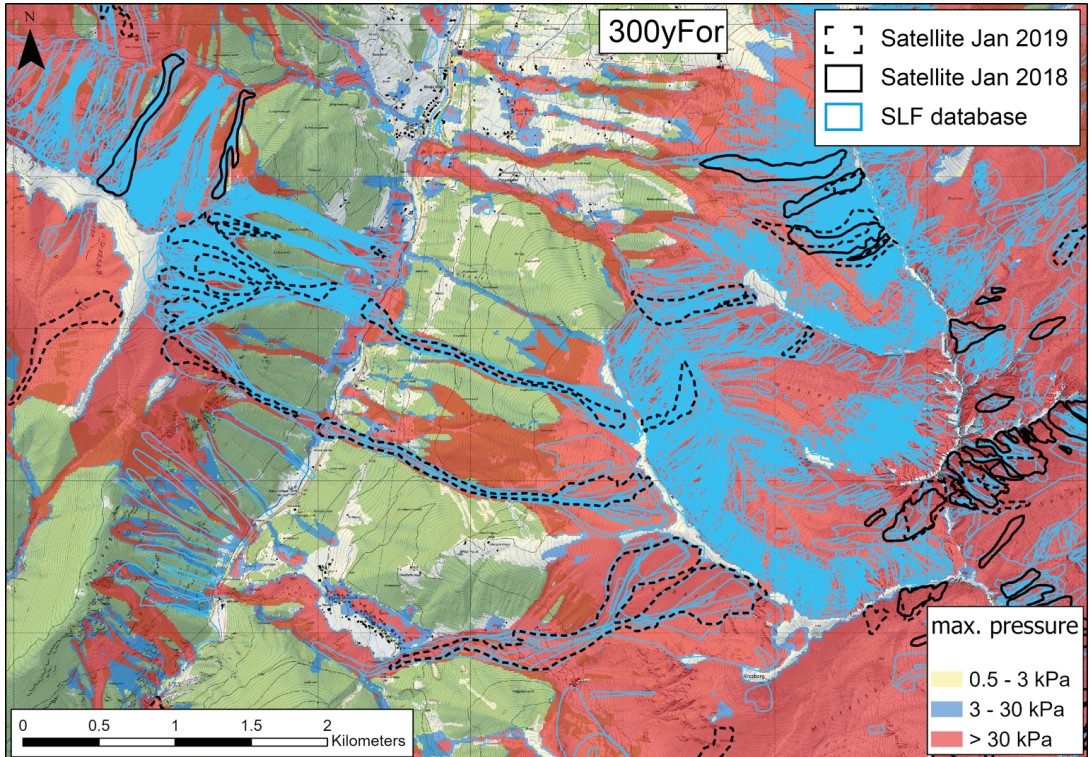

**Figure 11: Scenario 300yFor overlay of the mapped avalanches from the different data sources (satellite mapping and SLF database) for the region south of Davos. Only very small areas of even the rarest events are not captured by the hazard indication map. For example, in forested slopes in the southeast, where the forest structure might have changed, since the recording of the avalanche event (map source: Swiss Federal Office of Topography)**

## 6 Discussion

### 6.1 Strength and weaknesses

The results of the large-scale simulations capture the area endangered by rare avalanches (100 – 300y return period) well. By providing spatial continuous information, this product sets a new level of detail and accuracy for hazard information outside the zones captured by the official hazard maps (90 % of the area in the canton of Grisons). This enables new applications for practitioners (section 6.2). By simulating four different return period scenarios (10y, 30y, 100y and 300y) with and without taking the protective effects of the forest into account, detailed hazard analysis for different problem definitions can be performed. Furthermore, the presented approach can be applied to regions worldwide, where a high spatial resolution DTM, information on the protective effects of present forest and information on extreme snow depth increase is available. We already successfully applied this approach to the regions Trentino, Livigno, Aosta and Langtaufers in Italy (Monti et al., 2018; Maggioni et al., 2018; Bühler et al., 2018a) as well as regions in Canada (Sykes et al., 2021), Alaska, India, Afghanistan, and New Zealand. An adapted version of this approach was applied to simulate typical skier avalanches over entire Switzerland (Harvey et al., 2018). Currently we are simulation the entire canton of Valais (5'224 km$^2$), the second largest Alpine canton of Switzerland after Grisons. These applications demonstrate the big potential of the presented approach for automated hazard indication mapping. As the processing chain is automated, it enables reproducible hazard indication mapping applicable in mountain regions around the world, where the required input data is available.



However, as the presented approach is fully automated and performed by computers, assumption have to be made (section 3.6). These assumptions and further restrictions of the presented approach leads to important limitations that have to be considered when interpreting the results. First of all, not the entire area that is not affected by the simulations (even at the very rare scenario 300yFor) is completely save. The runout of very extreme avalanches, releasing in large bowls as a whole and where the large snow volumes confluence in a gully, can be underestimated. On the other hand, the runout of very small release areas can be overestimated. The impact of powder avalanches as well as the effects of protection buildings and the occurrence of several avalanche events within the same avalanche track are neglected. These limitations on runout have been detected by the involved avalanche experts of the canton and SLF at isolated examples. The overall assessment of the simulated runout revealed a very good agreement with the expert assessments (section 5.1) and only very few cases were observed, where the avalanche runout was underestimated (section 5.2). The automated delineation of the avalanche release areas (PRA) sometimes produces polygons that would be delineated differently by the experts. Considering these limitations, the simulated runouts from these PRA agree again mostly with the assessment of the experts.

Secondly the protective effects of the forest is dynamic and has a considerable influence on avalanche release in the Alps (Bebi et al., 2021; Bebi et al., 2001; Teich et al., 2012). The simulations can only consider the state of the forest at a specific point in time and are not yet able to incorporate forest changes. Together with the neglection of present mitigation measures, this leads to the main deviations of the simulations to the validation data. For the interpretation of the results, potential future changes of the protective effects of the forest have to be considered. The presented tool enables the assessment of future forest conditions on avalanche hazard.

## 6.2 Applications in practice

As 90 % of the area within the canton, potentially endangered by snow avalanches, is not covered by official hazard maps, the hazard indication maps produced in this study are the base to assess hazard within these uncovered regions. For example, this is important, if mountain huts or cabins are planned from scratch or extensions are requested. Also, if expansions of ski resorts or further alpine infrastructure is planned, the simulations are a first point of reference. The produced indication maps are applied as reference scenarios for the first assessment of potential hazard impacts. They enable the generation of risk indication maps as performed already for the Rhaetian Railway (RhB) within selected regions.

By comparing the simulations to the existing hazard maps, potential shortcomings, wrong or changed assumptions or changes in the boundary conditions of the hazard maps (e.g. forest and terrain changes) can be identified. Older hazard maps can be assessed and verified. However, the consideration of mitigation measures causes a large part of the deviations between hazard maps and the presented simulations (section 3.6), this has to be considered for this comparison. Nevertheless, if areas with high damage potential show considerable avalanche impacts in our simulations but are not yet covered by official hazard maps, they can be identified and carefully checked with more detailed hazard assessments.

As all scenarios were calculated with and without taking the protective effects of the forest into account (section 3.5), these scenarios can be applied to visualize, quantify and evaluate the protective effects of the forest. The delineation of protection forest is today based on the simulations performed in the project SilvaProtect-CH in 2004 (Gruber and Baltensweiler, 2004; Gruber and Bartelt, 2007; BAFU, 2013) applying a 25 m digital elevation model, a different PRA algorithm (Maggioni and Gruber, 2003), the avalanche dynamics model AVAL-2D (a precursor of the RAMMS model) and the forest information that was available at that time. Therefore, we expect major differences in the delineation of the protection forest. Because of the high economical and societal relevance of the protection forest functions in Grisons (Stritih et al., 2021), changes in the delineation of protection forest have very large impacts. Therefore, more time and discussions are needed to implement the



new simulations for this application. The work to do so has already started. Taking the effect of climate change, which is more distinct in alpine regions, into account, the presented scenarios serve as a baseline to assess potential changes in the future.

Last but not least the produced spatial continuous hazard indication maps prove to be a valuable tool to communicate and discuss risk between different stakeholders, including the population of alpine communities and decision-making authorities. Concluding, the presented maps are now an important base for the spatial continuous assessment of potential avalanche danger and allow for an efficient prioritization for in-depth hazard assessments such as hazard maps within the canton of Grisons.

## 7 Conclusions and outlook

In this study we present an automated approach to generate hazard indication maps for snow avalanches for the entire canton of Grisons, with an area of 7'105 km$^2$ the largest state of Switzerland. Four different scenarios (10y, 30y, 100y and 300y) are calculated, covering the whole range from relatively frequent to very rare events. A remote sensing-based identification of forests with protecting capacity against avalanche releases is developed. In total, approximately two million individual avalanche release areas (PRA) are delineated, loaded with estimated snow volumes per scenario and simulated with the state-of-the-art avalanche dynamic model RAMMS. The resulting spatial continuous hazard indication maps are then validated with existing, legally binding hazard maps, by expert assessment and by outlines of avalanches mapped with optical satellite data. The main deviations are caused by changes in the forest structure, the underestimation of very rare avalanche releases in large bowls and inaccurate mapping within the reference datasets. However, as the datasets applied for validation do only cover scenarios with undefined return periods and not all of the most extreme avalanche events possible, the validity is limited (even though they are the best datasets available). Some room for improvement we identify by the implementation of the protection measures in the avalanche release areas. But this is a very difficult task, as the protective effect for specific areas and different scenarios have to be defined. This is already a very delicate issue in the hazard mapping process for individual tracks and it is very hard to automate meaningfully.

The product is already applied in daily practice by the cantonal authorities for approximately one year and proves to be a valuable tool for:

- initial hazard assessments
- the evaluation of infrastructure planning in areas not captured by the official hazard maps (90 % of the area in the canton of Grisons)
- the evaluation and adaption of existing hazard maps
- the delineation and rating of protection forests
- the communication and discussion of avalanche risks with stakeholders.

The reproducible, spatial continuous maps enable efficient prioritizing of further investigations and also serve as "second opinion" for assessments performed by avalanche experts. Such maps can be calculated for any area where high spatial resolution digital elevation models (DEM), information on the snow climatology and the protective effects of the forest are available. Today, such information can be deducted from satellite data (Sykes et al., 2021). Any available avalanche observations in the region are very helpful to check the validity of the results.



In the future when better input and validation datasets get available, for example summer DTM from high-density LiDAR point clouds or even snow covered DSMs derived from airplanes (Bühler et al., 2015; Meyer and Skiles, 2019) or satellites (Marti et al., 2016; Deschamps-Berger et al., 2020; Eberhard et al., 2021), the existing approach can be refined with updated and partly more detailed terrain, snow cover and forest information. To assess potential effects of climate change on avalanche

hazard we plan to implement RAMMS::EXTENDED (Bartelt et al., 2012; Buser and Bartelt, 2015; Bartelt et al., 2015) into the processing chain, allowing for the consideration of snow erosion, the temperature of the snow layers, the generation of free water and therefore the transition of flow regimes. This may have large impacts on avalanche runout and allows for considering specific avalanche types such as powder or wet snow avalanches. This would also enable the spatial continuous analysis of climate effects on the scale of states or even entire countries. The presented approach forms thus the base for future

developments in automated hazard indication mapping.

**Acknowledgment**

The authors want to thank the Department for Forest and Natural Hazards AWN of the canton for Grisons and the Office for Civil Protection (ABS) of Liechtenstein for the mandate to perform this work and for their valuable contributions. The authors are grateful for the various contributions from the SLF staff and internal discussions. This enabled us to digitize a lot of SLF

know-how and experience. They thank Christian Ginzler, Lars Waser from the Swiss Federal Institute for Forest, Snow and Landscape Research WSL and Adrian Zaugg (SLF) for their contributions to the forest layers. They also want to thank Fabiano Monti from AlpSolut, Livigno, Margherita Maggioni, University of Torino and Christoph Oberschmied, civil protection Alto Adige for their cooperation with LSHIM in Italy.

**Author contributions**

YB developed the processing chain and wrote the manuscript with contributions from all co-authors. PB and GS developed the classification for the protective effects of the forest. MC developed the RAMMS::LSHIM model. AS and AC contributed to the simulation part. CM defined the climatic regions and calculated the ΔHS3 to derive the d0. SM and LS were the leading avalanche experts from SLF, RK the leading expert from the canton of Grisons and SW, the leading expert from Liechtenstein, evaluating the results and proposing various improvements that were incorporated.

**Data availability**

The PRA and the simulations results presented in this study will be published on EVIDAT (https://www.envidat.ch), the WSL data portal, upon the final publication of this manuscript.

**Competing interests**

The authors declare that they have no conflict of interest.




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
