# Peer review of "Automated avalanche hazard indication mapping on state wide scale"

_Natural Hazards and Earth System Sciences, 2022_

## Author Response (AR1)

**Reviewer #1**

Here are our answers to your suggestions (your comments in italic):

• Page 1, line 13-14: For the canton of GR, data from the "SilvaProtect" project are available at the hazard indication map level. However, these only show potential areas with avalanches. In contrast to the maps presented in this paper, the data from SilvaProtect do not show impact parameters such as avalanche pressure.

We clarify this point in the revised version by explicitly mentioning the products of our approach. We added the following text (in red):

For a large part of the area (90% in the case of the Canton of Grisons) only strongly generalized hazard indication maps are available (SilvaProtect), not showing impact information such as pressure.

• Page 2, lines 1-6: The avalanches in SilvaProtect were originally modeled for the delineation of protective forests. It was not the goal to use it to create a hazard indication map. However, due to the lack of alternatives, the data is sometimes used to identify potential areas with avalanches.

We add the following statement to this section:

A generalized version of these results is still applied in Switzerland today, due to a lack of alternatives, resulting in hazard indication maps delineating the vast majority of alpine terrain a hazard area, lacking detailed information.

• Figure 1: In this figure, the climatic regions appear for the first time. It would be helpful to explain briefly that these regions are used to regionalize avalanche modeling.

We will add this information to the Figure caption:

Figure 1: Map of Eastern Switzerland visualizing the extent of the canton Grisons (black outline), the major towns (red stars) and the largest ski-resorts (blue stars) as well as the climatic regions (dashed lines) applied in section **Fehler! Verweisquelle konnte nicht gefunden werden.** to regionalize the release depth (d0) of the avalanches (map source: Swiss Federal Office of Topography)

• Page 4, lines 12-13: The standard procedure for snow avalanche hazard mapping in Switzerland defined by the Federal Office for the Environment FOEN defines only three different return periods 30, 100, 300 years. The return period 10 years is optional.

We will add this information:

Based on the standard procedure for snow avalanche hazard mapping in Switzerland defined by the Federal Office for the Environment FOEN in collaboration with SLF, we simulate the four return periods 10, 30, 100 and 300 years, where the 10 years return period is optional in the guidelines.

• Chapter 3.2: This chapter is short. It is not possible to understand how the effect of the forest was assessed. Threshold values are mentioned without presenting

them. In order to be able to understand the results, the limit values should be presented or it should be shown at the beginning of the section in which paper the limit values can be found (Bebi et al. 2021).

We extended this section considerably by adding more details, in particular about the different threshold definitions for "gap-threshold", "forest cover-threshold" and the calculation of terrain roughness. We also refer – in the beginning of the section and later – to Bebi et al. (2021) where the assessment and delineation of the protection forest was more in the focus and where a comprehensive overview of the variables, threshold values and definitions for delineating avalanche protection forest is given (Table 2 in Bebi et al. 2021) *Page 5, lines 7-8: "The main input datasets are the binary forest information (chapter 3.2) and the digital terrain model (DTM, chapter 3.1)."*

The calculation of the gap threshold considers regional differences in snow depth distribution according to Margreth (2007) and compensates for underestimation of tree heights by the used vegetation height model. The "forest cover-threshold" takes into account the coverage at different spatial scales and optimizes for the detection of trees, especially in a critical range between 3 and 5 m (see details in Table 2 in Bebi et al. (2021)). In order to account for increased protection forest capacities of high terrain roughness compared to smooth surfaces, we delineate areas with a high surface roughness with the "Vector Ruggedness Measure" (VRM) according to Sappington et al. (2007) based on the 2-m DTM (SwissAlti3D) and a moving window of  $5 \times 5$  m. Areas with a value > 0.02 and no lateral convex curvature are considered as rough (Brožová et al., 2021).

• Page 5, lines 7-8: "The main input datasets are the binary forest information (chapter 3.2) and the digital terrain model (DTM, chapter 3.1)."

We will add the missing subsection numbering:

information (chapter Fehler! Verweisquelle konnte nicht gefunden werden.) and the digital terrain model (DTM, chapter Fehler! Verweisquelle konnte nicht gefunden werden.)

• Chapter 4.1: This chapter explains the identification of protection forests. In Switzerland, the cantons are obliged to delineate protection forests. Therefore, a comparison with the existing protection forest delimitation of the canton GR would be interesting.

We agree that a comparison with the existing protection forest layer of the canton is interesting. However, we classify all forest present based on remote sensing input and not only the forest estimated as official protection forest by the responsible persons. We are planning to start a further study, comparing our new approach with existing layers in collaboration with the canton of Grisions. However, in our opinion it would overload this paper to include comparisons already here as it would also be necessary to explain and discuss the comparison.

• Figure 7: The colors shown for maximum pressure lead to misinterpretation. In hazard maps, the colors express the hazard level and not the maximum pressure. We recommend using the colors of the intensity maps (three different greens) to display the maximum pressure.

We had intensive discussions within the author team about the optimal coloring of the maps. We see your point of proposing other colors for the pressure values. However, as we want to compare our results to hazards maps, we decided to use the colors corresponding to the threshold values used in hazard mapping in Switzerland (< 3 kPa, 3

- 30 kPa and > 30 kPa). In Figure 10, where we use the same coloring as in Figure 7, it is essential to do so to enable a comparison of our results with the existing hazard maps. As this is a scientific paper targeting scientists, we think this coloring is most meaningful. Based on these arguments we want to keep the current color scale of Figure 7.

• Chapter 5.1: For the comparison of the existing hazard map with the modeled ones, it is important to know how the protection forest is considered in the existing hazard maps. Please explain briefly so that the comparison can be understood.

Thanks for this hint, we will add a brief explanation of how the forest is considered in hazard mapping in Switzerland:

Only dense closed forests covering the entire release area provide very good avalanche protection. When drawing up a hazard map, it is assumed that no avalanches will start there. If there are openings in the forest or if the forest is scattered, avalanche release can no longer be excluded and the protective effect is reduced. If avalanches finally release above a forest, it can be destroyed. Such forest stands or forest stands in the avalanche track are not taken into account in hazard maps. can be destroyed. Such forest stands or forest stands or forest stands in the avalanche track are not taken into account in hazard maps.

• Figure 10: Both maps show the colors red, blue and yellow, but the colors do not have the same meaning. In hazard maps, the colors express the hazard level. In the avalanche hazard indication map the colors show the maximum pressure. The hazard maps show not only the flow avalanches but also the powder avalanches. In the indication map, only the flow avalanche is taken into account.

As explained in the previous point on the color scale, we choose the colors according to the threshold values applied in Switzerland for hazard mapping. This enables in our opinion the best comparison between the hazard maps and our results. We will add a sentence highlighting again that we do not take powder pressures into account to make it clearer.

• Chapter 5.2: In the modeling only one type of avalanches was modeled (dense flowing part of dry avalanches). In the event analyses, other types of avalanches were probably also recorded. How does this affect the comparison of real data vs. model?

The avalanches mapped by satellite and present in the existing cadaster are in the vast majority of the cases the outlines of the deposited debris. Therefore, this corresponds well with our simulations. There might be some isolated cases, where also impact of powder clouds were mapped. But we do not expect that this has any influence of the statistics. We add a sentence to clarify this point by adding the following text:

The avalanches mapped by satellite and present in the existing cadaster are, in the vast majority of the cases, the outlines of the deposited debris. Therefore, this corresponds well with our simulations. There might be some isolated cases, where also the areas impacted by powder clouds were mapped. But we do not expect that this has noteworthy influence on the statistics.

Reviewer #2

Dear anonymous reviewer. Thank you for reading our manuscript. Unfortunately, your points are quite unspecific. For us it is hard to improve the manuscript based on your suggestions. In particular we cannot understand how you come to the conclusion of major revisions for our manuscript.

Please find our comments to your points here:

• Abstract. No quantitative result is found here.

We add quantitative results (number of simulated avalanches, percentage of affected area per scenario) to the abstract:

We calculate eight different scenarios with return periods ranging from frequent to very rare as well as with and without taking the protective effects of the forest into account, resulting in a total of approximately two million individual avalanche simulations. This approach combines the automated delineation of potential release areas, the calculation of release depths and the numerical simulation of the avalanche dynamics. We find that between 47 % (most frequent scenario) and 67 % (most extreme scenario) of the cantonal area can be affected by avalanches. Without forest, approximately 20 % more area would be endangered.

• Introduction. Might you let me the objective(s) of this study?

The last section describes, in our opinion, the objectives of this study precisely:

We apply this newly developed automated hazard indication mapping tool to the canton of Grisons in Switzerland, validate the results with existing hazard maps, assessments of avalanche experts, historical avalanche cadastres and avalanche records mapped by satellites. Then we discuss the strength and weaknesses of this approach and provide an outlook on future applications and potential improvements.

• Methods. The part may be revised to be greatly concise.

In our opinion this part is already concise. However, went through the text carefully to further improve it.

• Results and validation. Table 1 is not easy to understand. Does "2'056" mean "2 056" or "2.056"? Same questions exist in Table 3, page 15.

The format of numbers in tables is given by Copernicus. We will stick to this template.

• Conclusions. The part is also needed to be simplified. For example, in this part a conclusion is only concluded from your own study and the thus conclusions from references, e.g., the sentence in line 36, page 18, "Today, such information can be deducted from satellite data (Sykes et al., 2021)", may be erased.

The Last part of the conclusions is the outlook on ongoing and future improvements. Therefore, it is important to include this reference here.